# The role of cooking practices in the transmission of the foodborne parasite *Taenia solium*: A qualitative study in an endemic area of Southern Tanzania

**Karen Schou Møller**[1]*, **Pascal Magnussen**[2], **Stig Milan Thamsborg**[1]*, **Sarah Gabriël**[3], **Helena Ngowi**[4], **Jeanette Magne**[5]

1 Department of Veterinary and Animal Sciences, Faculty of Health and Medical Sciences, University of Copenhagen, Frederiksberg C, Denmark, 2 Department of Immunology and Microbiology, Centre for Medical Parasitology, Faculty of Health and Medical Sciences, University of Copenhagen, Copenhagen N, Denmark, 3 Department of Translational Physiology, Infectiology and Public Health, Faculty of Veterinary Medicine, Ghent University, Merelbeke, Belgium, 4 Department of Veterinary Medicine and Public Health, College of Veterinary Medicine and Biomedical Sciences, Sokoine University of Agriculture, Morogoro, Tanzania, 5 Danish School of Education, Campus Emdrup, Copenhagen NV, Denmark

* smt@sund.ku.dk (SMT); karenm@sund.ku.dk (KSM)

**Data Availability Statement:** All relevant data are within the paper. All study participants have been assured complete anonymity and that their interviews or demographic data will not be shared

## Abstract

The pork tapeworm *Taenia solium* is a zoonotic food-borne parasite endemic in many developing countries causing human cysticercosis and taeniosis as well as porcine cysticercosis. It mainly affects the health of rural smallholder pig farmers and their communities, resulting in lower health status, reduced pork quality, and economic loss due to condemnation of pigs or low pricing of pork. This qualitative study aimed to identify key food related practices linked to consumption of pork at village level, of importance for transmission of taeniosis. We used an interpretivist-constructivist paradigm in a multiple case study of exploratory qualitative research design. Data was acquired through guided and probing interviews with 64 pork cooks, and 14 direct observations in four villages in a *T. solium* endemic area of Mbeya Region in the Southern Highlands of Tanzania. The study showed that the informants were members of communities of practice through their pork cooking practices, one community of practice for the restaurant cooks and one for the home cooks, learning, sharing, and distributing their cooking skills. Furthermore, the analysis showed that the pork cooks generally had some awareness of there being something undesirable in raw pork, but they had very diverse understandings of what it was, or of its potential harm. Major potential transmission points were identified in restaurants and in home kitchens. It appears that the pork cooks act according to socio-cultural and economic factors guiding them in their actions, including pressure from customers in restaurants, the family values of tradition in the home kitchens, and the culturally guided risk perception and appraisal. These practices might generate potential transmission points. Future research on interventions aimed at preventing the spread of *T. solium* taeniosis should recognise the importance of tradition and culture in risky food practices.

with anyone outside the group of authors. Sharing data in a data repository could lead to potential recognition of study villages and/or study participants e.g. through age, gender, familial relationships, or descriptions indicating locality, which would violate the agreement that the study participants consented to prior to engagement in the study. Therefore, we have shared anonymised excerpts within the manuscript, as per PLOS One Guidelines for Qualitative Data.

**Funding:** This work was funded by the European & Developing Countries Clinical Trials Partnership (EDCTP; grant number DRIA2014-308), project titled: "Evaluation of an antibody detecting point-of-care test for the diagnosis of T. solium taeniosis and (neuro)cysticercosis in communities and primary care settings of highly endemic, resource-poor areas in Tanzania and Zambia, including training of – and technology transfer to the Regional Reference Laboratory and health centers (SOLID)"; and the German Federal Ministry of Education and Research (BMBF; grant number: 01KA1617). The funders had no role in study design, data collection and analysis, decision to publish, or preparation of the manuscript.

**Competing interests:** The authors have declared that no competing interests exist.

# Introduction

This study focuses on cultural and culinary aspects of transmission of the pork tapeworm, *Taenia solium*, which is an endemic parasite found in most pig-raising developing countries of Africa [1], the Americas [2] and Asia [3]. *Taenia solium* causes three disease entities, where two are caused by the parasitic larval stage (metacestode) lodging in the muscles or CNS of either humans (human cysticercosis) or pigs (porcine cysticercosis) forming macroscopic cysts, and the third is due to the adult tapeworm in the intestine of humans (taeniosis) [4]. In 2010, World Health Organization identified cysticercosis as one of the Neglected Tropical Diseases and a leading cause of death from food-borne disease [5]. Fig 1 shows the life cycle of the parasite, where eggs are shed by tapeworm-infected humans through faeces and passed on to humans or pigs causing cysticercosis. If pork with cysts is eaten by humans, the parasite will become an adult worm in the small intestine shedding eggs and the life cycle is complete. Humans ingesting eggs may develop cysticercosis, in particular neurocysticercosis, which can cause severe, chronic headache and epilepsy [4, 6].

A cross-disciplinary One Health approach involving both the agricultural sector (including the veterinary sector) and the human health sector is essential in control of the parasite in sub-Saharan Africa [8]. This approach should include the social sciences as the interactions between the human host, the porcine host and the parasite take place in a cultural context as well as a biological and physical one [9]. However, to date no large-scale, government run control programmes using a One Health approach have been implemented [8]. In Tanzania, and particularly the region of Mbeya, there is a relatively high prevalence of taeniosis according to the literature review by Braae and coworkers [10] and Mwanjali and coworkers [11], who identified a local taeniosis prevalence of 4.1%. Health education of farmers is prerequisite to obtain control of infections, and this was investigated in Northern Tanzania [12], where the authors found good uptake in knowledge, but not a significant change in practices. This was also the case in Mexico, where Sarti et al. [13] found that knowledge of *T. solium* was increased through education, but that the change in practices was not correlated to knowledge uptake. This underlines the need for a more socio-ethnographic practice-based research approach in order to explain the underlying reasons for the transmission of *T. solium* as a part of the control strategy for taeniosis.

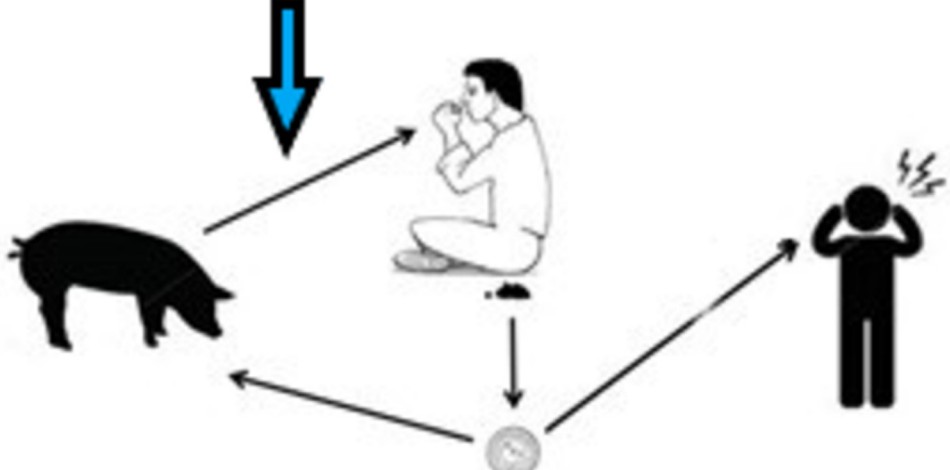

**Fig 1. The (simplified) life cycle of *Taenia solium*, modified from** [7]. *T. solium* eggs are shed by infected humans and passed on to humans or pigs through ingestion of faecal matter, causing (neuro)cysticercosis. If infected pork is consumed, humans can develop taeniosis, thus completing the cycle.

Mwidunda, Carabin [14] conducted an intervention study assessing knowledge and attitudes towards taeniosis and cysticercosis among school-children in northern Tanzania and found an uptake in knowledge following health education. They used qualitative methods to explore the post-intervention attitudes of eating infected pork and found that the participants were more likely to consume the pork if it was thoroughly cooked, but that intervention also resulted in increased acceptance of (attitude towards) selling infected pork instead of condemning it. This supports the findings by Boa and coworkers [15] showing that infected pork is not being condemned but is sold via alternative channels. Thys and colleagues [16] showed that eating undercooked pork was perceived as a health threat in communities in Eastern Zambia, but called for further research on why and in what situations people are infected. Ngowi and coworkers [17] found that the vast majority of interviewed villagers in the Southern Highland of Tanzania (same catchment area as the present study) would consume known infected pork, but the study did not describe practices in handling the infected pork nor cooking and concluded that future studies should include the individual's situation as well as the community's capacity in order to enhance behavioural change. Previous research shows that the actual cooking practices resulting in consumption of undercooked pork are not looked into despite thorough cooking being considered a crucial part in controlling the transmission of the *T. solium* taeniosis [18] and thus has failed to produce data on exactly when and where the transmission from pork to human happens (blue arrow in Fig 1).

To address this knowledge gap, the present study aimed to investigate the situations when cooks are preparing meals, both in restaurant kitchens and at home. Based on analysis of qualitative data from field studies in four rural Tanzanian villages, the study describes cooking practices and pork consumption and identifies potential situations where transmission of *T. solium* (leading to taeniosis) may take place.

The primary research question was: What are the main potential transmission points of *T. solium* taeniosis in rural kitchens of Southern Highlands of Tanzania?

Secondary research questions were: What role do the social and cultural practices of the pork cooks play in transmission of taeniosis? What guides the pork cooks in their everyday choices in regards to pork handling and cooking?

## Transmission

The study draws on two main concepts in transmission–direct/indirect transmission and transmission through cross-contamination.

The term transmission of infection covers the process of passing a pathogen from one host to another. Direct transmission occurs through direct contact between hosts while indirect transmission involves an environmental phase or phase in an intermediate host. In this context transferring of the infective larvae (metacestodes) to a person is through the oral route during consumption of infected and undercooked pork [19]. However, transmission through cross-contamination, defined here as the inadvertent transfer of the metacestodes from infected raw or undercooked pork to a tool, plate, hand or workstation and further on to the ready-made meal via unsanitary handling procedures, may potentially also take place. Unwashed hands and tools in the kitchen during cooking may represent a threat of cross-contamination [20, 21].

Undercooking of pork is considered an important factor in transmission and is central to this study. Møller and colleagues [22] recently showed that cooking for 10 minutes, reaching a core temperature of 80˚C is needed to effectively kill all *T. solium* metacestodes in pork. Fried and shortly cooked dishes not reaching this core temperature may potentially lead to transmission of viable metacestodes.

## Practice theory and communities of practice

To analyse the practices that the pork cooks apply in their daily cooking and handling of pork, the study draws on the theory of practice by Schatzki, who defines practice as a "temporally evolving, open-ended set of doings and sayings linked by practical understandings, rules, teleo-affective structure, and general understandings" [23]. In the context of pork cooks, the practices inherent in pork cooking and handling are viewed as an unlimited set of temporally unfolding organised actions. They are connected in time and space with a common causality and intentional direction [24]. According to Loscher and coworkers [25], these *doings and sayings* are tied to a particular practice by the practical and general understandings of that practice. The cooks know how to react in situations within the practice of cooking and possess "the bodily know-how and implicit knowledge to conduct, recognise, and react to other activities" [25 (p. 4)]. The cooks have an understanding of the values and aesthetics of the cooking practice, "including a cultural and societal sense of appropriateness and rightness" [25 (p. 5)]. In essence, people tend to do what makes sense for them to do in any particular situation, based on their experiences (individually learned or socially inherited). Within this frame of analysis, the actions of the pork cooks in this study are described and analysed to offer a potential explanation of the transmission of *T. solium*.

The study also draws on theories of Communities of Practice (CoPs) [26] and Situated Learning [27, 28]. These offer explanations about how people learn, centered on social and cultural aspects of such processes. They define learning as a social skill and specify that learning occurs through participation in practice, in a particular situation within the community (Fig 2). The theories also claim that "*learning, thinking, and knowing are relations among people engaged in (an) activity in, with, and arising from the socially and culturally structured world. This world is itself socially constituted.*" [27]. Dreier [29] argued that practice happens in

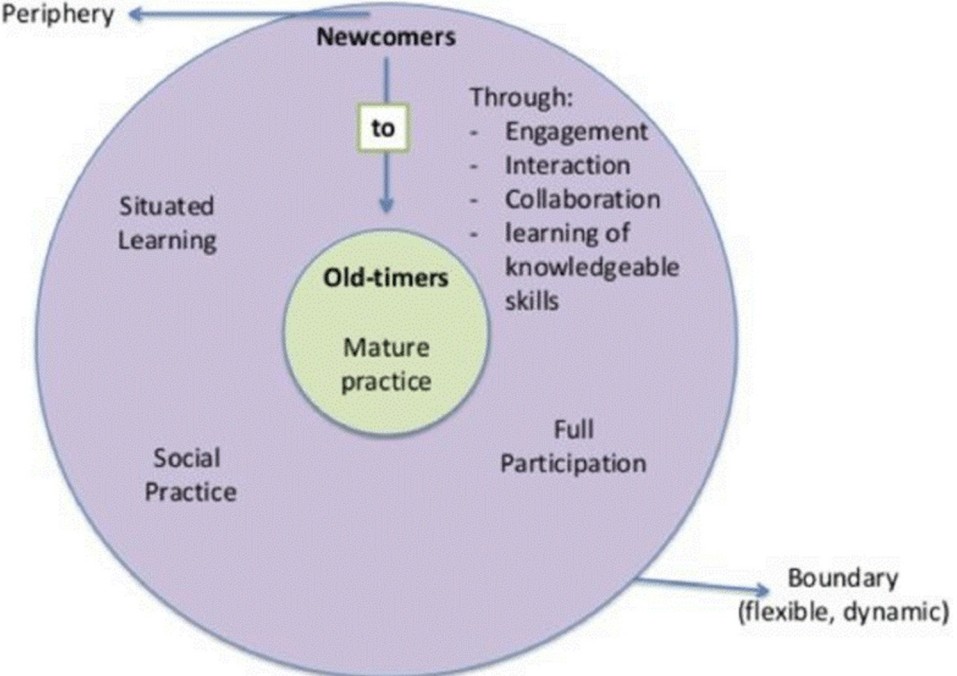

**Fig 2. The flow of the CoP.** The entry of newcomers in the communities of practice (CoPs), learning by engagement and interaction. The newcomers continue their path into the centre of the CoP, becoming old-timers at the practice (modified from [28]).

motion and that tracing the movements of a person was necessary to understand the actual practice of that person. Therefore, the observations of this study were carried out by spending a day with each home cook informant, thereby including the different practices that precedes the actual practice of cooking. This allowed us to form a relational bond with the woman and through that observe and participate in her normal practice best possible. The restaurant cooks were each traced for a minimum of 3 hours up to a day, allowing us to gain insight into their usual practice of cooking and selling pork.

The theory of learning has been used to study how newcomers enter a community of practice, for instance a professional community or a place of work, as *legitimate peripheral participants* [28], what constitutes participants' roles and tasks in a given community [30] and how new participants develop their skills over time [31]. In this study, CoP theory is used to analyse how the cooks learned to cook pork and how they pass on this knowledge to the next generation of cooks; the home cooks to the children in their care and the restaurant cooks to people who wish to join the professional cooking community.

## Method

### Study design and methods

The study design was a multiple case study of transmission factors in cooking practices. It was carried out as a community-based participatory ethnographic research project. The project consisted of a 5-month (July-November 2018) fieldwork in four rural villages of Mbeya Region in the Southern Highlands of Tanzania. The study used qualitative observation and interview methods, thus applying the interpretivist-constructivist paradigm [32] as central in trying to understand the role of cooking practices in transmission of taeniosis through a case study of the pork cooks [33]. The research was therefore not neutral, because the researcher cannot be a neutral figure in the observation, but must interpret what is observed in practice [34]. This non-neutrality begins when electing what cases to include and choosing what subjects to elaborate on in the interviews. Within the constructivist approach, meanings are constructed through experience and through the use of material resources [35]. The reconstruction of the practices of cooking a pork meal as a joined experience between researcher and cook allowed for the researcher to observe the cooks' creation of meaning within the practice of real-time cooking of the pork meal.

### Research context

Mbeya region, covering 35,954 km$^2$, has more than 280,000 pigs and holds 22% of the national pig herd [36]. Pork consumption is very common as community data show that 88–92% of the inhabitants in the region eat pork [11, 37, 38]. To identify study villages, a list of all villages in Mbeya District Council (total of 942) was produced. Villages that were known to have been involved in *T. solium* research previously were excluded, as these villagers would have been sensitised in their knowledge of *T. solium* and thus not be representative for the general rural population of the area. Villages inaccessible by car in the rainy season were excluded, as were villages more than 4 hours' drive from Mbeya City. Out of the remaining 122 villages, four were randomly selected. However, two villages did not want to participate when approached, so two others were selected based on the advice of the Regional Veterinary Officer in Mbeya Region. The four villages were typical rural villages of this region, with the implication that they were communities of 700–1100 inhabitants of predominantly Christian belief, they were inhabited predominantly by subsistence farmers and otherwise fulfilled two major criteria: pork was consumed and sold in the village and based on their willingness to participate. The four villages were named Peak Village (Peak), Lowland Village (Low), Factory Village (Fac)

**Table 1. Characteristics and tribal ethnicity of 64 informants from the four study villages.**

| | Peak Village | | | | Mountain Village | | | | Lowland Village | | | | Factory Village | | | | Total |
|---|---|---|---|---|---|---|---|---|---|---|---|---|---|---|---|---|---|
| | YW | OW | RC | | YW | OW | RC | | YW | OW | RC | | YW | OW | RC | | |
| Tribes | | | MC | FC | | | MC | FC | | | MC | FC | | | MC | FC | |
| Safwa | 6 | 5 | 1 | | | | | | | 3 | 3 | 1 | | | | | 19 |
| Nyakyusa | | 1 | | | | | | | 4 | 2 | | | 2 | 2 | | 3 | 14 |
| Ndali | | | | | 6 | 3 | 3 | | | | | | 1 | 1 | | | 14 |
| Malila | | | | | 1 | 3 | 1 | | | 1 | | | | 1 | | | 7 |
| Other | | | | | | | | | 2 | | | | 4 | 2 | 1 | 1 | 10 |
| Total | | 13 | | | | 17 | | | | 16 | | | | 18 | | | 64 |

YW: Female home cooks of 18–39 years of age, OW: Female home cooks of 40 years of age and older, RC: restaurant cooks, MC: Male restaurant cooks, FC: Female restaurant cooks.

and Mountain Village (Mou) based on geospatial or landmark location to secure anonymity. The four villages were not chosen nor named based on any known differences of practice.

**Characteristics of study population and data collection.** The study informants consisted of pork cooks divided into three groups: restaurant cooks (RC) of either sex (FC or MC), female home cooks of 18–39 years of age (YW), and female home cooks of 40 years of age and older (OW). The dividing of gender was imposed as it is usually (if not always) women who cook at home, and mostly (though not all) men cooking as a profession. The division on age at home was chosen based on the assumption that older women might cook differently from younger women.

In each village, approximately six (1–7) persons in each group were identified with help from the village chairperson, based on both purposive and snowball sampling (Table 1). These numbers were estimated as preliminary, but adjusted according to presence of informants, willingness to participate etc. The informants were coded for anonymity within the group by assigning a number to every person. This produced a unique identification code for each informant resembling Low-YW-03 or Fac-MC-01 and so forth. The majority of the informants were of the Safwa tribe (Table 1), originating in the area of Mbeya City. They are traditionalists and often believe in both traditional healers and acknowledged religions. The second most common tribal ethnicity was the Nyakyusa tribe, which originates from the Tukuyu area South of Mbeya. The traditional Nyakyusa people are agriculturalists often keeping larger amounts of milk-producing cattle and thus tend to include milk in their diets and cooking. The Ndali tribe originates from the southernmost areas of Tanzania, bordering Malawi and Zambia east and south of the study area, and the Malila tribe originates in the Northern Mbeya Region. The remaining tribal ethnicities originate from other areas of the country.

A total of 64 interviews with cooks were performed and 14 direct observations were made of cooking situations, during which informal conversations took place. Cooking sessions with home cooks (all female) were participatory and took place in the homes of the cooks (Table 2); restaurant cooks were observed when working at the restaurant or at their selling point, according to Table 3. Furthermore, four interviews with slaughterers (who were also restaurant cooks) were conducted, as was one observation of a slaughtering session with the local slaughterer in a private village slaughter slab in Lowland Village. The observations and the informal talk during slaughter focused on the meat handling practices of the slaughter, and the fate of the various parts of the animal after slaughter.

In the following, the term *slaughterer* is used for a person actually performing the slaughtering, where the term *butcher* is used for a person selling the meat in a designated meat shop, a butcher's shop.

**Table 2. Overview of number of interviews.**

| VILLAGE | INTERVIEWS | | | | TOTAL |
|---|---|---|---|---|---|
| | Restaurant cooks/slaughterers | | Female home cooks | | |
| | Male | Female | 18–39 years | ≥40 years | |
| LOWLAND | 3 | 1 | 6 | 6 | 16 |
| PEAK | 1 | 0 | 6 | 6 | 13 |
| MOUNTAIN | 4 | 0 | 7 | 6 | 17 |
| FACTORY | 1 | 4 | 7 | 6 | 18 |
| TOTAL | 9 | 5 | 26 | 24 | 64 |

Throughout Tanzania, there is a tendency of pork restaurant cooks being male, which was echoed in three of the study villages (Mountain, Peak and Lowland Village). In Factory Village, however, there were clearly more female restaurant cooks in the village, which the gender distribution of participants from this village also reveals. We did not find any explanation for this.

Further data sources consisted of rich handwritten field notes from all interviews and all informal talks (at observations and at walk-and-talks) including drawings of particular situations or places, and photos of locations, infrastructure, water sources, village landmarks, or particular situations.

Guided interviews were conducted in Kiswahili through a female Kiswahili/English translator who was also fluent in several tribal languages. The PI of the study also spoke Kiswahili ensuring the accuracy of the translation. The interviews consisted of open- and closed-ended questions producing qualitative data as well as basic demographic data. This allowed the interviewer to probe and explore interesting points until saturation. The main questions included information on the recipe of the preferred pork dish, the preferred choice of meat and reasons behind, as well the sensory, cultural and traditional aspects guiding these choices. Furthermore, we explored the informants' experience with "white nodules" (the chosen description of cysts) in pork along with their understanding of origin and implications.

To ensure the confidentiality of the informants of the present study, the authors cannot share the data forming background of the study. However, whenever data are needed for the transparency and validation of a finding, quotes and excerpts from interviews and informal talks have been included in the manuscript. In such cases, the informant has been anonymised.

**Table 3. Overview of number of observations of cooking sessions involving pork preparation.**

| Village | Observations | | | | Total |
|---|---|---|---|---|---|
| | Restaurant cooks | | Female home cooks | | |
| | Male | Female | 18–39 years | >40 years | |
| Lowland | 3 | 1 | 2 | 3 | 9 |
| Peak | 0 | 0 | 0 | 0 | 0 |
| Mountain | 0 | 0 | 0 | 0 | 0 |
| Factory | 1 | 4 | 0 | 0 | 5 |
| Total | 4 | 5 | 2 | 3 | 14 |

Lowland Village was chosen as the Focus Village and thus where the home observations took place. Mountain Village did not cook pork commercially at the time of visits due to severe drop in pig population because of a recent African Swine Fever outbreak. Peak Village had not slaughtered pigs at the times of visits, so no observations were carried out there.

## Data management and analysis

Forty-eight interviews were audio-recorded, transcribed and translated into English. Another sixteen interviews were audio-recorded and thematically translated to ensure saturation within the selected themes–in total 64 interviews. In one instance (in the interview with Low-FC-02), the technical equipment did not function properly and the interview was not audio-recorded. The information from this interview was shortly after the interview collected in writing based on field notes and memory, and included in the data set. The four slaughterers interviewed were also pork cooks and thus included in the 64 interviews, but the interviews were guided into the field of pig slaughtering focusing on the distribution of the various parts of the pig post-slaughter. There was one observation (lasting 2 hours) of a pig slaughter during which informal talk took place. The observation of and informal talk during the pig slaughter were not audio-recorded, but thorough field notes were written and photos were taken to aid the analysis of this session. Furthermore, informal talk took place naturally when spending time with the informants, either shopping for ingredients with the home cooks prior to the cooking session, walking with informants (or other members of the community) from one interview to another, spending time in various places in the villages, or observing daily life from the town square. Handwritten field notes were written and transcribed every evening or early the next day.

All written data were coded and structured using Nvivo Qualitative Analysis Computer Software 12. This added to the consistency of the qualitative data analysis process, because it allowed the researchers to see patterns in the handling and cooking of pork and to connect these patterns to the social and cultural practices. For example, when the informants were asked of their preferences of meat cuts when buying at the butcher shop, their response was separated into "Colour", "Texture" and "Bones/fat/skin" depending on what they focused on. When asked "Why that piece?" their responses were separated into "Taste", "Beliefs of superiority", "Health" or "Teeth problems". Often the answer would be a combination of these reasons and thus coded in all the appropriate categories. The informants' focus on the visual appearance of the meat and their attraction towards certain meats and their repulsion towards other kinds, as well as the reasons why, were recorded. This aided the analytical process in the categorisation of "visual attractiveness of meat". The transmission points were identified by combining the pre-existing research on the viability of *T. solium* cysts and cooking temperatures of pork [22], and on the restaurant cooking practices [38].

Each guided interview was performed using only pen and paper, but immediately after the interview additional information about the informant (including basic demographic data), the interview situation, or the atmosphere was noted and all handwritten field notes were transcribed into a computer at the end of each day of fieldwork. Furthermore, a photo was taken consensually of each informant.

All non-written data aided in the analysis of practices and situations by jolting the memory and by reviving the ambiance of the given setting. Furthermore, photos and sketch drawings also played an important role in analysing the physical setting of the cooking practices and life around them.

## Ethical considerations

Ethical approval for this study was obtained from Tanzania Commission for Science and Technology (permit number 2019-039-NA-RCA-2018-29). The Regional Veterinary Officer and the appropriate District Veterinary Officers all approved of the research study and were included in the planning and implementation. All informants were above 18 years old and provided written consent. In case of illiteracy, the informant was read the consent form in the

**Table 4. Answers to the question "Have you ever heard of "white nodules" in pork?" from 63 informants.**

| Ever heard of white nodules in pork? | Peak | | | | Mountain | | | | Lowland | | | | Factory | | | | Total (%) |
|---|---|---|---|---|---|---|---|---|---|---|---|---|---|---|---|---|---|
| | YW | OW | PC | | YW | OW | PC | | YW | OW | PC | | YW | OW | PC | | |
| | | | MC | FC | | | MC | FC | | | MC | FC | | | MC | FC | |
| Yes, and has some correct knowledge of what happens if eaten | 1 | 1 | | | 4 | 3 | 1 | | 3 | 1 | 1 | | 2 | 3 | | 1 | 21 (33) |
| Yes, but do not know what happened if eaten | 5 | 4 | 1 | | 2 | 1 | 3 | | 3 | 5 | 2 | | 2 | 2 | 1 | 3 | 34 (54) |
| No, never | | 1 | | | 1 | 2 | | | | | | | 3 | 1 | | | 8 (13) |
| Total | 6 | 6 | 1 | 0 | 7 | 6 | 4 | 0 | 6 | 6 | 3 | 0 | 7 | 6 | 1 | 4 | 63 (100) |

Note that one interview (Low-FC-02) is missing due to technical difficulties, therefore a total of 63.

presence of a literate witness, and signed with a thumbprint. The witness also signed for the legitimacy in these cases.

## Findings

The study identified a number of factors of potential risk for transmission in relation to the handling, cooking and consumption of pork in the study area. These were categorised into the *site* where potential transmission can occur (at home or in a restaurant), and the *practices* that potentially increase the risk of transmission. Each site contained various combinations of practices and the findings are therefore presented by site of potential transmission and the practices involved are discussed in relation to this site.

As a background for understanding factors of potential transmission, we first discuss the cooks' awareness of *T. solium* and taeniosis.

### General awareness of the risk of taeniosis amongst cooks

During interviews it became clear that the vast majority of the all the informants (both restaurant and home cooks) had heard of or seen white nodules in pork (55 out of 63 informants (87%) (missing information from one informant)) (Table 4). Of the 55 informants who had seen or heard of white nodules, 21 (38%) had an explanation when asked if they knew what happened if one ate meat with the white nodules in it, rendering 34 (62%) informants without any explanation–they had just seen or heard of the "white nodules".

All the restaurant cooks had some awareness of the "white nodules" in pork, although the vast majority of them did not know what happens if it is eaten (Table 5). In the group of the home cooks, the majority knew *of* the "white nodules", but could not answer what happens if eaten. There was no difference between older and younger home cooks in either reply (Table 5).

**Table 5. Answers to the question "Have you ever heard of "white nodules" in pork?" from 63 informants, as in Table 4 but divided into informant groups.**

| Ever heard of white nodules in pork? | YW | OW | RC | Total |
|---|---|---|---|---|
| Yes and has some correct knowledge of what happens if eaten (%) | 10 (38) | 8 (33) | 3 (23) | 21 |
| Yes, but do not know what happens if eaten (%) | 12 (46) | 12 (50) | 10 (79) | 34 |
| No, never | 4 (15) | 4 (17) | 0 (0) | 8 |
| Total | 26 | 24 | 13 | 63 |

Note that one interview is missing due to technical difficulties.

There were names for the white nodules in all of the local languages, which was translated from the Kiswahili word "*wadudu*" into "bugs". The majority of the informants in the "Yes, but do not know what they are" group did know that the nodules were "bugs" or even "worms", but not more than that. An older woman in Factory Village explained:

"*They are worms. They come from the dust where the pig lives.*" Fac-OW-05

This woman knew only that they were worms, but had no correct knowledge of anything else pertaining to the parasite. This was very common in the group of informants with some knowledge.

Of the 21 informants with an explanation as to what would happen if the white nodules were ingested, none revealed appropriate knowledge of the result of eating infected pork. Some of the informants with knowledge of the white nodules had quite elaborate knowledge of the parasite, but often only parts of it were correct and often important parts of the knowledge on implications of infection were missing. As an older woman from Peak Village explained:

"*Yes, I know them [the white nodules]. They are harmful to pigs but not to humans. They inspect for them under the tongue, they look like peas, but they have white mucus inside. The pigs get it from eating a certain kind of grass with human faeces on.*" Peak-OW-03

She knew very well what the white nodules were and had some correct knowledge of the parasite, but expressed it was not dangerous to humans. Interestingly, most informants did not seem to doubt their knowledge of the white nodules; they knew what they were and knew of the implications of them. They did not seem to think they only possessed *some* knowledge.

Another informant explains:

"*They are just normal bugs, they don't do anything. We used to eat them all the time at home.*" Low-YW-06

This excerpt shows that the woman has full confidence in her knowledge and does not consider the white nodules to be a problem, since they used to eat them in her childhood home and apparently, nothing happened. These type of replies were common from the group of people who knew that the white nodules were a bug or worm.

Other explanations of the implications of ingesting the white nodules were confused with other illnesses and thus the informants would explain symptoms pertaining to other illnesses, such as "*the worm will only have an effect on the people who do not take frequent baths if they walk barefoot*" (Peak-OW-05). The year before the study there had been a severe outbreak of African Swine Fever, so several informants connected the symptoms from that disease (e.g. weakness, weight loss, reddening of the skin) to the white nodules seen in pigs. Some of the explanations were "*They make the pig weak*" (Moun-OW-03) and "*They make the pig skinny and red and even the meat looks funny*" (Moun-OW-02). Other explanations were inexplicable, such as "*the bugs will slowly eat the human flesh*" (Moun-OW-06), or "*If you eat it you will become like a pig*" (Peak-OW-06). All explanations had the common trait of being delivered with great confidence and conviction. Interestingly, in Factory Village the cooks (restaurant and home) were the only informants to know boiling the meat before frying or to boil it for long time to kill potential nuisances. As one younger woman in Factory Village said:

"*They are bugs. When I see these bugs, I know that [the] pig had a disease and that is why we are advised to boil the meat for a long time so that the bugs can die.*" Fac-YW-04

The village communities all had health inspectors, but in this village particularly, they had been warned to cook the meat in order to kill the parasite. This did not appear to be the case in any of the other villages.

Several informants would disclose that they had in fact eaten infected pork or experienced other people eating it, and that it would cause a popping sound in the mouth from the cysts exploding. A young woman from Peak Village explained:

> "*They are very white and they usually stay in the steak of the meat, not on the fat, and when you cook and eat them they pop in the mouth.*" Peak-YW-03

(Here, it should be noted that all informants used the term "steak" as a general term for meat without fat, bones, tendons etc.) The informants frequently described this sound or sensation of the cysts popping in the mouth or while cooking, which they tended to dislike. A male restaurant cook explained that he would be ridiculed by the customers if he served it, which deterred him from buying and cooking it. He explains:

> "*It disgusts people, but it does not do anything. It just makes people laugh at you when they hear the kachakachakacha sound.*" Moun-MC-03

The findings show that although most cooks have some form of knowledge about the visible cysts, their knowledge was not necessarily applied in their cooking practices. When the cooks seemingly have such confidence in possessing the right knowledge, it could counteract changes in their behaviour. However, this needs further research.

## Sites for potential transmission

Restaurants and homes were the two major sites for pork consumption and are thus the sites where the practices are analysed. Market places were an additional site where pork was eaten ("mobile restaurants") and where transmission therefore was possible, so these are included in the analysis of restaurants.

**Restaurants.** In the study villages, a pork restaurant constituted a pork-eating venue as well as a butcher shop or meat shop, as previously described by Kimbi [39]. Restaurants were situated in the village centre or market square and only sold pork. They were open on days where there had been a pig available to slaughter (or money to buy the pig to slaughter) and thus not every day. They are here termed "fixed restaurants".

However, a restaurant was not necessarily tied to a specific location–it 'followed' the cook. In the study villages, there were daily small local markets offering vegetables, fish, and small cooking and household items, upon availability. Once a week, there was a bigger market in a larger village nearby gathering all the small markets in one place. Pork cooks from the small villages would travel to the larger market to sell pork dishes to the increased number of potential customers. In the case of Lowland Village, the female cook (Low-FC-02) would make her pork soup and carry it on her head walking to the market place every Wednesday. This type of mobile restaurant was often seen in the markets in the region.

**Practices of cooking and handling pork in restaurants.** In the fixed pork restaurants were small piles of fried meat on a large plate alongside bigger piles of raw meat. The cooked meat pieces were approximately 5 cm cubes, had been dipped into the boiling oil to remove the blood from the surface of the meat (a practice called dipping). This dipping practice lasted less than a minute and was done to make the meat more visually attractive. The practice was observed at all participating restaurants selling fried pork. After dipping, the meat was put in a large dish to drain the excess oil. Customers would either buy the dipped meat or buy raw

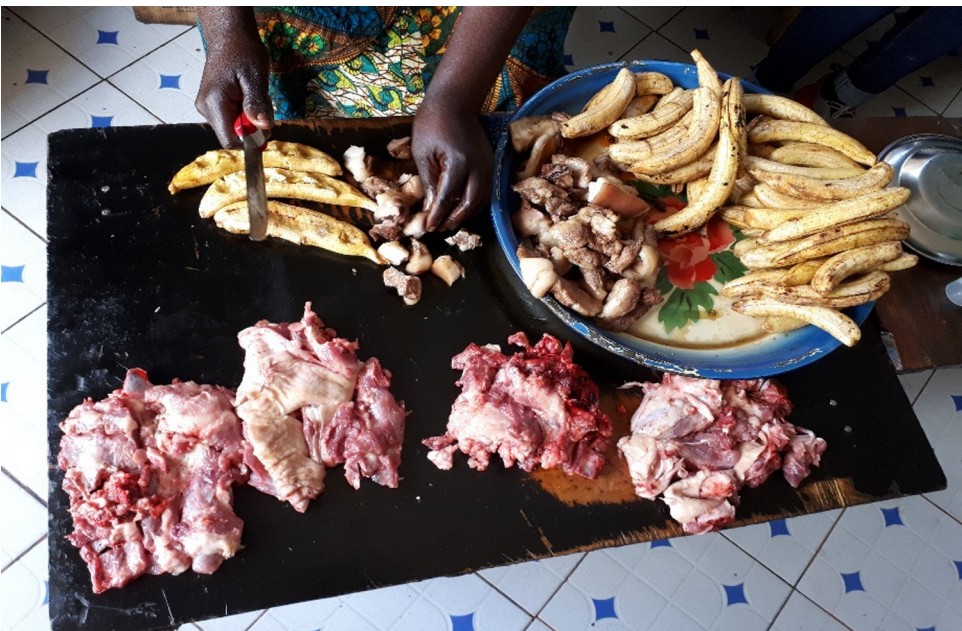

**Fig 3. A restaurant workstation in Factory Village showing the three types of pork for sale in the restaurant.** On the right on the plate, plantains are seen alongside the dipped pork. On the left, the restaurant female cook is cutting the ready-to-serve pork and plantains on the wooden table where also the raw pork is located (in front).

meat to take home. The raw pork pieces were displayed according to price and some negotiation was observed, because the cooks would decide what meat, fat and/or bone pieces to include in each pile. Only in one restaurant, a weight scale was used (Low-MC-01). The eat-in restaurant guests would order from the dipped meat and the cook would further cook the meat in oil until they deemed it thoroughly cooked. The pork was in most cases served with fried plantains (Fig 3).

Based on observations in the restaurants, hand- and tool hygiene arose as a major issue for further investigation. The restaurant cooks were cutting the dipped and ready-to-serve pork and plantain directly on the table without cutting board or safe distance to the raw meat. This practice can be seen on Fig 3.

The restaurant cooks frequently would use the same knife for cutting raw and ready-to-serve meat. There were in total nine restaurant observations and not one cook washed or rinsed knife or hands during observations. The workstation would be superficially wiped with a cloth or apron to remove small raw meat pieces on the cutting area. There was water for hand washing present in all restaurants, but this would not be used to clean the workstation.

A point for potential transmission was observed when the cooks used the same plate to transport raw meat from the workstation to the fireplace, and then serving the meal to the customer after cooking. This practice of same-plate serving was observed in three out of nine restaurants.

All interviewed restaurant cooks knew not to serve undercooked pork, but few knew the reason for this (See Table 4). They knew that cooking it thoroughly was important because of health reasons and some even had explanations including parasites, but generally they had very different ideas of what was thorough cooking and how to measure it. During observations, one male restaurant cook explained he knew the pork was sufficiently cooked once it was "*floating on the oil*" with bubbles around the edges combined with his sensation of time passed (Low-MC-01). Another restaurant cook (Fac-FC-04) explained at observations that she

"just knew" when the pork was done, but she could not explain why or how–only that she had cooked pork many, many times and therefore knew. This sense of "just knowing" was common at the observations in restaurants (and, in fact, in the homes too). Some mentioned the colour of the meat playing a small role, but most restaurant cooks "just knew" when the meat was thoroughly cooked. This showed the embodied knowing as something the cooks had learned through years of practice and repetition and therefore had gained a feeling—an understanding—of when the pork was thoroughly cooked. It was generally difficult or even impossible for them to explain this or provide the rationale behind their practices. None of the restaurant cooks used a thermometer, nor did they on any observation use some sort of time taking device.

**Practices of cooking and handling pork in homes.** The home was a key site for preparation of pork and thus practices related to preparation and serving of pork in the home were important to analyse in the context of transmission.

Through interviews and observations, it became apparent that the dishes served in the home differed from the dishes served in the restaurants. Where the restaurants all served pieces of pork deep-fried in oil and served with fried plantain or chips, or a pork soup with plantains, the vast majority of the home cooks would make a thick pork stew with ugali (a common maize staple in sub-Saharan Africa consisting of maize flour cooked with water to form a stiff porridge). The younger home cooks would cook a plain thick stew with tomatoes and oil, where the older home cooks often would prefer to cook traditional stews from their tribal area (e.g. with plantains, potatoes, or cassava, as thick stews or more soup-like). At all observations of cooking sessions with the younger women, a thick stew was the recipe of choice. A few home cooks would explain cooking fried pork for their families but only on very rare occasions such as a good market day where the economic situation allowed this. One woman said in her interview:

> "*I always make stew with it [the meat], because the fried meat only, then it is not enough for the family*"

> Fac-YW-04

This excerpt did not stand alone and shows that the home cooks were cooking the stewed pork because it would stretch the meat compared to other recipes and thus feed the whole family.

During observations in the homes, there were situations that could potentially favour contamination. The women used the same knife to cut raw meat and vegetables and the same pot for washing raw meat and vegetables, albeit sometimes rinsing with water prior to use. However, in two of the five home cooking sessions, the women also served a side salad with raw vegetables washed in the same pot as the raw meat and cut with the same knife as the raw meat. In these situations, cross-contamination would be possible, as the cysts could transfer from the raw meat to the raw vegetables. The use of a cutting board was not observed, as all the women would cut meat and vegetables in their hands. Hands or knives were not washed between tasks, but wiped in a piece of cloth or in an apron.

At home, practices differ greatly compared to the restaurants. We found three main practices related to degree of heating/cooking meat in the homes that could be essential for potential transmission. These three practices were intertwined and a situation could contain several practices at the same time.

**Uncertain cooking time.** The first practice of interest was the practice of *estimating cooking time*. The women were typically cooking a thick pork stew, as mentioned earlier. They

would buy the raw pork at the restaurant and cut it up in pieces. These pieces were washed in water and transferred to the cooking pot using hands without wiping off remaining water. Some women would add a bit of water, some would not. The meat would fry in its own fat and the excess water. When this water dried out, the meat was typically put aside on a plate. In the same pot, the sauce would be made using tomatoes, onion and sometimes, other vegetables. The sauce would dry out quite a bit, leaving the desired thick stew. Then the meat was returned to the pot and heated, and the stew was thought to be ready. This 'estimation' of cooking time was an embodied practice that the women had learned through their CoP as women cooking at home. It was not just one embodied practice of cooking but also a whole perspective of practices of cooking, where the women *do* cooking. Several women expressed during interviews and observations that they knew that the meat was well cooked, when the water was dried up. This would not be a precise estimation, as the amount of water in a dish would vary. There was no particular time-taking or measurements of readiness of the meat present at any of the cooking sessions. One woman described the estimation as follows:

> "*I soft dry [the pork]. It still has its whitish colour; I just fry it very little. (. . .) I just check on the meat if it is cooked according to my satisfaction, then I take it out of the stove. But estimating time, I don't know.*"

Low-YW-03

This woman used the colour of the meat to estimate the level of cooking. Several of the women explained how they just knew when the pork was ready, but were not able to give an estimated cooking time–just like the woman in the excerpt above. Most women estimated the level of cooking based on a combination of: drying of water, the colour of the outside of the meat, the firmness of the pork pieces, her sensation of time passed, and her experience, in order to create a safe estimate of the readiness of the pork. Some women would bite into the pork piece, and return the remainder of the piece to the pot, if she was not satisfied. When the younger home cooks in Lowland Village cooked the stew during the participatory observations they cooked the pork pieces for more than 30 minutes, one for as long as an hour and a half. However, one must remember that they were cooking with us as guests, so it might not reflect the actual cooking time of an everyday meal.

The various practices of estimating the cooking of the pork had one thing in common–they did not have a distinct reference of time passed, but were all embodied tacit knowledge emerging from the practice of cooking. They had been learned socially in the CoP from when the women were young girls through repetition and later reproduction of the practices of "doing cooking".

**Lightly cooking of meat.** The second relevant practice was revealed during several interviews, informal talks and cooking sessions, where the women would describe how they would only lightly cook the meat. There were multiple reasons to fry only lightly and again, these could be intertwined.

When pork pieces fry in oil for a long time, they will shrink in size and become hard and dry in texture. The women would often cook the pork pieces only lightly to preserve the soft texture.

Several of the woman described how this was important for them in order to cook food that babies and people with bad teeth could eat. One older woman described this in her interview:

> "*I like soft meat because I have problems with my teeth*" Peak-OW-04

This was a very common response to the question of whether they preferred soft cooked pork or dry cooked pork. The desire to preserve the soft structure of the meat was also important in families with children, as the babies could not eat hard fried pork. The woman expressed cooking soft pieces for the children:

"*I don't dry it, because if I dry it my kids will not be able to eat*" Peak-YW-07

This was a common response from the women cooking at home. The children with only a few teeth had to be able to chew the pork with their gums and thus the meat had to be soft.

Another reason to cook the pork pieces only lightly proved to be of an economic nature. In order to make sure that the meat purchased was enough to satisfy the members of the household, the women cooked only lightly to prevent the pieces from diminishing through drying. One woman explains:

"*The reason we like lightly stewed meat it is because we want the food to be enough and satisfy kids. Also small children they can't eat dry meat*" Low-YW-01

In an informal talk with an older woman living with her husband and six grandchildren (Low-OW-01), she explained how she would buy a small amount of meat and cut it into seven pieces–one bigger piece and six smaller pieces. The bigger piece was for her husband and the smaller pieces were for the children–she herself would just eat the stew with leftover scraps of meat and bone. She would only fry the pieces lightly, because neither the husband nor the small children had the teeth status for chewing dry meat–and they wanted the pork pieces to be as big as possible. This shows that cooks could engage in the practice of lightly cooking the meat for both reasons–to preserve the size and preserve the softness.

**Fast cooking.** The last of the practices important to mention is *fast cooking*. Many of the women expressed a desire or need to cook fast. One woman explained in a cooking session that she normally cooked with charcoal, but when she was in a hurry, she would use the firewood stove, because the liquids would dry faster. Therefore, she knew the dish would be finished sooner. She also explained:

"*You know, most people are in a hurry when they cook they don't care much if the meat is cooked. We don't have a constant time of boiling the meat. Maybe when we have time to settle and cook. When we are tired from the farm works, we are starving so we are in a hurry to eat, so time it is not an option*" Muv-YW-01

She also mentioned the fact that there might be time to "settle and cook", but that this was not the case in her everyday life. Here, she had to produce a meal fast in order to satisfy the family's urgent needs. Another woman explained how she cooks "with ugali", meaning cooking the thick stew and serving it with ugali:

"*Most of the time I cook pork for lunch to eat with ugali because it is the fast dish or we order the meat which is already cooked from the restaurant. (. . .) Because it is faster*" Fac-YW-02

This illuminates the need for cooking fast and why the women tend to cook the stew and not cook other, more time-consuming dishes. In many situations, the three main practices mentioned here were combined and the younger woman would lightly cook the pork pieces in a thick stew to produce a fast meal after a long days' work in the farm.

**A community of practice under change.** In the traditional CoP, the newcomers are the younger generation learning from the older. Indeed, the women cooking at home all learned from either their mother or another female relative. Only one woman explained that she did not have a mother nor had she lived in a family, so she learned from God and by looking at other people (Peak-OW-01). When asked, many of the women seemed surprised to even be asked of who taught them how to cook–the CoP of cooking was so innate and inherent to them that it seemed strange even to talk about it.

One younger woman from Lowland Village explained the CoP in her childhood like this– with a little grin—when asked who taught her how to cook:

> "*My mothers at home, of course, and groups of people like on the wedding ceremonies. I just watch how they cook and I learn by watching them*" (sic) Low-YW-03

The term "mothers" refer to the cultural construction of the Tanzanian community of women and the familial construction of the older generation of women in the family. In Tanzania, if the father has more than one wife, the ones senior to your birth-mother are called "big mother" (in Kiswahili: *mama mkubwa)* and the junior ones are called "little mother" (in Kiswahili: *mama mdogo)*. Furthermore, the term *Mama* is used as a sign of respect for women who have had children or who have reached a certain age, regardless if they are sisters or aunts. In fact, when a woman becomes a mother, her calling name changes to Mama followed by the name of her oldest child (for example Mama Joseph). This stays with her the remainder of her life and indicates the importance of children. After you have a child, you no longer keep your own individual identity, but gain the respected title of mother. The woman from the excerpt grew up among several mothers and thus learned how to cook from them. Women of different families meet and cook the meals for ceremonies (like weddings and funerals) and the woman in the excerpt also learned from watching these women cook together. The CoP in cooking is formed by these groups of women within the family and extends to the larger groups of women when families meet and cook together. This gathering of women from different families—cooking together, watching each other, learning from each other, add to the influences into the CoP. This is the traditional way of the CoP, from newcomer to old-timer.

Findings showed that there had in fact been a change in cooking practices in general through the last generation. Fig 4 shows the CoP under change with the influx from its members and their everyday lives. In the following, the different types of influx will be elaborated.

Several older women explained how they learned to cook completely without oil, but either boiled the pork in water with salt or made a soup using groundnut, and potatoes or plantains. Soda ash (sodium carbonate, $Na_2CO_3$) was often added to the water to soften the meat. It is naturally occurring in the Tanzanian soil and is bought as a stone or as a powder in local markets. It neutralises the acids in the meat and breaks down the protein thus tenderising the meat while it is boiling. One older woman explained:

> "*My sisters taught me [to cook]. I was raised by my sisters since my mother died when I was very young. My sisters taught me to cook pork with salt and soda ash. The pork stew they also taught me but later on when they started cooking with tomatoes and onions*" Peak-OW-03

We did not observe the use of soda ash during the cooking sessions, but we did see the soda ash for sale on the local markets indicating its use in households. Another traditional cooking ingredient was groundnut powder. It thickens the stew or soup and adds flavour and the three older women in the Lowland Village were cooking pork soup with boiled plantains with groundnuts or a smoked pork stew (using pork strips smoked over the fire for 2–3 days) with

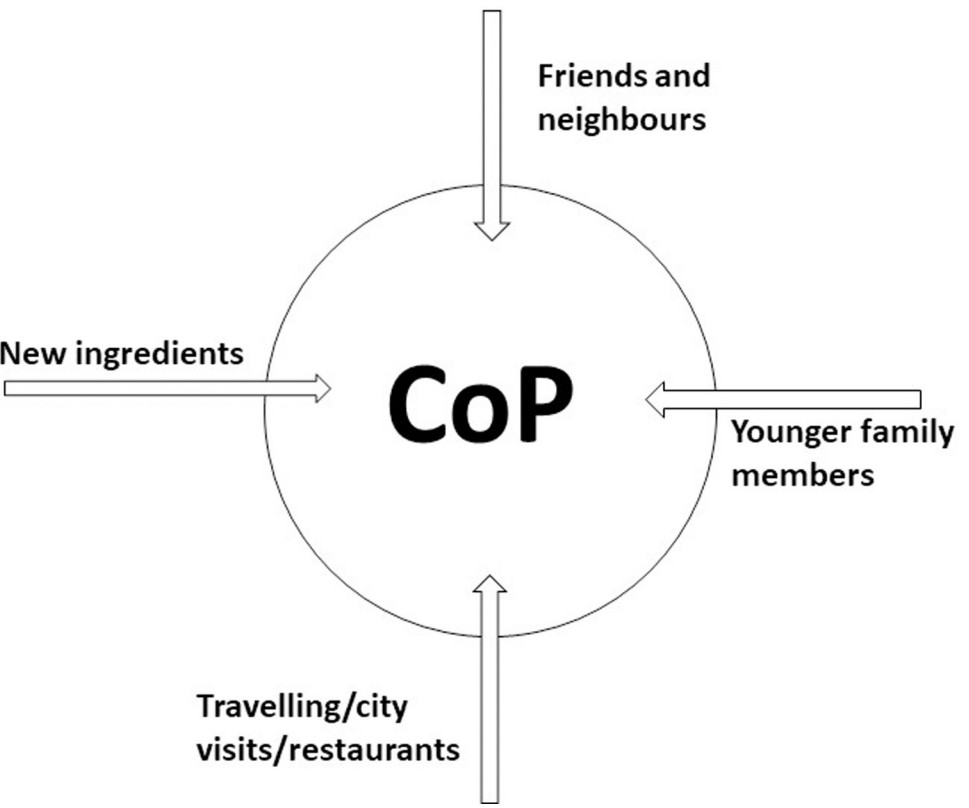

**Fig 4. The influx of new practices in an existing Community of Practice (CoP).**

groundnut powder during the observations of cooking sessions. They explained that groundnuts were a common ingredient in their childhood home cooking, but had largely disappeared from modern recipes. The home cooks explained how this was the way their mothers, grandmothers or other female relatives would cook when they were young and living at home. Learning in the CoP of home cooks traditionally went from the older women to the younger women but always through women. Modern cooking in the home, as represented by the younger cooks during cooking observations in Lowland Village, consisted mainly of the thick pork stew. The younger women explained how their mothers or an older female relative had taught them how to cook all foods, *except* this recipe. One young woman explained:

> "*My mother taught me to cook, but pork she boiled with salt. I learned to cook pork like this from looking at friends.*" Peak-YW-06

Even the older cooks had learned the recipe of the thick stew, but again, not from their mothers. They had learned this recipe from friends, neighbours, or their daughters. An older woman said her older children did not appreciate her traditional pork boiled in salt, so they taught her how to make the thick stew:

> "*[The children] were saying this meat doesn't taste good like the fried meat and it is too soft, I decided to change the way I cook. I decided to fry first before I make stew, since then they enjoy my cooking*"
>
> Moun-OW-02

An older woman in Lowland Village, who had learned to cook the thick stew from her daughter, backed this up. During an informal walk-and-talk of her garden, she explained how her daughter had cooked it for her, she liked it, and now it was her preferred pork dish (Low-OW-04). These are examples of the changes in the direction of the learning and reproducing within the CoP.

The restaurant cooks had learned to cook the fried pork sold at restaurants from watching other restaurant cooks, typically by eating in restaurants, or by watching friends, who were restaurant cooks. One female restaurant cook had learned from her husband, who died suddenly leaving her with the responsibility of providing for the family. She knew how to cook fried pork so she continued his business after his death. Several explained how they saw a market and thus a business potential in cooking deep-fried pork and started based on that. One key informant (Low-MC-01) explained how he did not eat meat at all, but still he was the leading pork cook and slaughterer in Lowland Village. He said:

"*For me what made me cook pork for business is the hardship of life, as youth when [you] reach a certain age you must find a way to get money. And I looked [at] what will be good for me to do. I started looking at my friends in town cooking, then I learned from them*". Low-MC-01

He had entered the trade because he had seen how it was becoming a steady trade in town while on a visit there, and decided to start his business in his own village–despite being a vegetarian himself. This indicates how the kind of trade is less important as long as it is economically profitable. This was the case for all the male restaurant cooks—they simply learned the trade out of necessity to fulfill the family's needs as heads of their household and not because of a desire for cooking. One female restaurant cook in Lowland Village (Low-FC-02) cooked pig's head soup and had started her business because the head is the cheapest part of the pig, but the rest of the female restaurant cooks sold fried pork like the male restaurant cooks. None of the restaurant cooks had any formal training, but all had entered the CoP due to the economic potential of the trade. None of the restaurant cooks mentioned having learned the recipe of fried pork from their mothers or female relatives. This is probably due to the fact that the fried pork is a modern recipe and that the community of practice of women cooked other traditional recipes.

It seems the element of travelling was also important to how the cooks learned new recipes. Several of the villagers explained how they travelled to town and tasted the new recipe of thick stew here. One young woman explained:

"*[I learned to cook the thick stew] at a restaurant in town.*" Low-YW-02

This was backed up by an older woman, who learned through her husband's travels:

"*My mother taught me how to cook the old-school way boiling and adding salt only. My husband told me how to cook the modern way of drying the meat. He saw it somewhere in town and told me how. Then I just tried.*" Moun-OW-04

Furthermore, several of the restaurant cooks explained how they had learned to cook in town at a restaurant. One male cook explained:

"*I was working in a restaurant in town and I learned by looking at the cook cooking the stew*" Fac-MC-05

These excerpts are examples of how travelling is an important element in the acquisition of new inputs into the CoP. The villagers of Mountain Village had to travel by foot to a nearby village for market day or onwards travel, but Factory Village and Lowland Village had daily minibuses passing through, connecting them with nearby towns. Albeit infrequent, uncomfortable and slow cheap daily transportation to bigger villages or towns makes it entirely possible to travel and thus the acquisition of new practices, recipes and ideas into the CoP of cooking continues. Travelling has also changed the availability of food, making rice available for even mountain residents although it grows in warmer climates of less altitude. In traditional cooking in the study area, they did not use many vegetables. An older woman explained:

*Woman: We didn't have tomatoes in those days, even to get an onion was a miracle, it was hard to get them. Interviewer: So the only ingredient you were using in those days was the salt?*

*Woman: Yes".* Moun-OW-02

This was an important finding as infrastructure and transportation is becoming more and more extended and thus the availability of ingredients will expand as well.

In conclusion, the findings reveal the main potential transmission points of *T. solium* taeniosis in the rural kitchens of Southern Highlands of Tanzania. They showed that the pork cooks would undercook pork in certain situations, and that the poor hygiene standards might play a role in transmission. Moreover, they showed that the pork cooks, both restaurant and home cooks, would act in a way that made sense to them in the particular situation. If that meant cooking lightly, in a different way or selling undercooked pork, then that is what they would do–because it made sense in that particular set of circumstances. They would be guided in their choice of practices by the particular circumstances in combination with the inherited and embodied knowing of how to cook pork.

## Discussion

The findings presented above give a picture of a CoP under change. Home cooks have learned cooking from their mothers, aunts and grandmothers by looking, trying, tasting and refining their skills and they in turn will teach their daughters and granddaughters. They are part of a practice-based community of women cooking and teaching each other [28]. These cooking practices are part of a larger assemblage of skills that women must learn and practice every day including child rearing, cleaning, fetching water and firewood, as well as some distinct pastoral duties such as sowing, harvesting and tending to goats and chicken [40]. We suggest that women engage in these duties from childhood and little by little, learn how to cook from more experienced members of the community. They are from an early age legitimate peripheral participants engaging in the CoP of women in rural communities.

The restaurant cooks learn in much the same way, albeit at a later time in life. They enter the CoP as adults, often due to an economic incentive, but they learn from the elder old-timers in the CoP, the people who have learned before them [28].

The CoP changes over time and an influx of new ideas or rationales can (and will) occur naturally. New ideas, e.g. recipes and food preparation practices, are introduced and thus the passing of skills from the younger generations to the older–opposite of the traditional inheritance of skills. In the future, with improving infrastructure, economy and means of transportation, the CoP will likely change at a faster pace due to the higher rate of influx into the CoP.

The change in the predominant recipe used in the home and at restaurants–from the slow-cooked pork soup or banana stew to fast-cooked thick stew at home or the even faster cooked

deep-fried pork pieces at restaurants–could be an important factor regarding transmission of taeniosis. Maridadi and coworkers [38] found that the vast majority of pork is eaten in the local restaurants and as the modern recipes do not necessarily entail sufficient cooking of the pork, this could pose a serious risk to public health in terms of increased transmission of *T. solium*. Moreover, there is currently no defined pork cooking time in the guidelines; the perception of well-prepared food is subject to the experiences of the cooks *while* they cook. The two younger home cooks would both cook the thick stew during observations, but the three older cooks cooked other more time-consuming dishes (soup with potatoes, thin stew with plantains, smoked pork stew with groundnut). Being older women, they did not tend to the farms far away and thus did not necessarily spend long time away from the house. This probably allowed them time for the cooking of slow-cooked dishes. The fact that the older women generally did not know how to cook the modern dishes could indicate that they did not need to cook fast in their younger days either–that modern times are busier and thus leaves less time for cooking. However, this area still needs to be fully researched.

The CoP of cooking in these rural Tanzanian villages has gone from traditional cooking to more modern cooking practices and the study found that one of the major potential sites in taeniosis transmission was the restaurants, where deep-fried pork was the most common dish and as described sometimes sold undercooked according to the customers' desires. This corresponds with the findings of Maridadi and colleagues [38], who found that the pork restaurants were locally known for serving undercooked pork, and when the majority of pork meals were consumed in the pork restaurants as opposed to home cooking, this would entail a major risk. This is furthermore corroborated by Kimbi [39], who found that the slaughtering in rural areas was very often conducted without official meat inspection and the meat was sold in restaurants to both dining customers and home cooks.

The young women cooking in the homes tend to cook for a shorter time compared to the traditional ways of cooking because of time constraints and exhaustion from farm labour. Urbanisation in rural areas of Mbeya Region has been noted in the most recent census [36]. As a result, the villages and towns become larger and thus the farming areas are not located by the villagers' houses as they traditionally were but further away, often on a rented plot. The families of the pork cooks often rented several small plots in different locations around the villages, so walking between these plots was strenuous and time consuming. It was also found that the economic situation of the home cook was an important factor as a reason for the home cooks to shorten preparation time pork. However, the study did not investigate whether this shortening of the cooking time did in fact lead to undercooking.

A commonality for all cooks in the study was that they practiced cooking based on what made sense in every particular situation. They combined the inherited knowing of cooking with what was at stake in any given situation. In essence, they did what made sense for them to do. This would change from situation to situation and even within one situation. There were a multitude of factors that were spilling into the practice, making it a collection of activities, an organised nexus of actions [23]. These are the "temporally evolving, open-ended set of doings and sayings linked by practical understandings, rules, teleo-affective structure, and general understandings" that is inherent in the Practice Theory [23]. Cooks will have to weigh potential long-term health implications against present-time socio-cultural factors such as their available economic resources and the need to feed their families, as long as infected meat is not excluded from consumption. When the money is scarce, feeding the family today becomes paramount and the distant future becomes less important. This was also found by Thys and coworkers [16] who in a study of eastern Zambian villagers found that villagers were aware of the risks, but were willing to take them for socio-economic reasons. These factors should be taken into consideration in the development of future interventions and governmental

policies, as they are significant for the complex assessment of risks relating to transmission. Furthermore, this is an area of research that needs more investigation to illuminate the factors important for risk appraisal and sense-making for the rural pork cooks.

When applying the ethnographic methods of observing, the researcher must always keep in mind that we can only observe the informant in a somewhat simulated situation. This means that the practices of the cooks during the observation session might be subjected to change *because* the cook is being observed. This is called the Hawthorne Effect and is a well-known bias of observation [41, 42]. It must therefore be questioned if the situations of the cooking sessions fully mimicked the actual situation of an everyday cooking session in terms of time elapsing. While the home cooks were only observed once for a day while cooking one meal, the restaurant cooks were observed cooking several meals, thus probably making the latter observed practices truer to the actual practices when not under observation. The observed practices might have been altered to increase the social desirability of the practice, and thus the practices that we *did* observe might represent the perceived desired minimal hygiene levels and safe cooking practices carried out. The Hawthorne effect might have masked several other not observed practices of enhanced transmission risk, but this needs further research.

There have been several attempts to control the transmission of the pork tapeworm [8]. However, they have not been context-based and tailor-made to support behavioural change at community level as shown by Ngowi and coworkers [17], who suggested including the community in the planning of future intervention programs in order to ensure the actual behavioural change needed for cessation of the transmission of *T. solium*. The present study supports these findings that knowledge *transfer* is not sufficient to change behaviour. We found that the cooks had basic knowledge of the existence of something making white nodules in pork, but that they did not know how the parasite impacts human or animal health, neither the seriousness of the impact, nor how transmission is prevented, corroborating with the findings by Maridadi and coworkers [38]. Currently, there are no guidelines for the day-to-day cooking practices on how to eliminate the risk of *T. solium* transmission disseminated to the cooks in the villages, but Møller and colleagues [22] found the viability curve for *T. solium* metacestodes when boiling infected pork. This could be included in future guidelines for safe cooking. In order to ensure the ethical soundness of future studies, these must also include a change in policy for meat control as well. The problem of infected meat is not only a matter of changing the behaviour of the people at the end of the line of consumption–it should indeed be a priority for policy makers to communicate the laws of meat inspection, hygiene regulations in the food chain, and, not least, ensure their effectuation and compliance. Despite governmental efforts imposing regulations that all slaughtered pigs must be inspected by a health official [43], meat inspection is still more or less non-existent in Tanzania, echoed by [11, 17, 39, 44]. The current official meat inspection guidelines were written in 1993 [43], and thus, an updated version might be relevant as well as enforcement of these guidelines. In constructing guidelines for safe pork cooking in educational intervention studies, it should be subject to debate how to make it ethically sustainable to ask the least resourceful to change their practice as it potentially would entail an economic burden for them. However, if the practices that potentially entail transmission are changed, so could the tradition. The practices learned within the CoP could change too and thus newcomers having learned the practice of boiling the pork before cooking it, for example, would eventually become old-timers and teach the new newcomers the new practice of boiling. This would take time, potentially even generations, but if the practice could become an integrated part of living, it would probably be more sustainable than an externally learned theoretical change in practice. However, this would require a change in the CoP of epidemiological research as well, as qualitative longitudinal intervention studies in rural populations are rare.

Including the Regional Veterinary Officer in Mbeya Region in the decision of which villages to include in the study could potentially be a selection bias. However, as the present study was of a qualitative nature, randomisation of the study villages was not essential to the study outcome. The most important features of the villages was the consumption of pork, the willingness to participate and share knowledge and practices, and the year-round accessibility by car. After careful explanation of the study to the Regional Veterinary Officer, he assisted by naming two villages, that would fulfill these features, and thus it was decided to include them, as long as the stipulated features were fulfilled. Furthermore, the acceptance and guidance from the government appointed regional officer-in-chief within this field and from the local Veterinary Inspectors is a prerequisite for any research in the region. This was achieved by collaborating in the selection of villages and this outweighed the risk of selection bias.

The findings of the study are probably representative for small pork producing communities of the region and even to other village communities of similar economic status outside the region. Many of the traditional, cultural and economic factors in this study are readily transferable to rural communities in other countries and thus the CoP under change could be too. Thys and coworkers [16] conducted a study based on focus groups and found that socio-economic and cultural practices played a dominant role in pig management, corroborating with Bardosh and colleagues [45] in their multidisciplinary study in Lao PDR. They found that the socio-cultural drivers in a rural village had a great impact on their intervention study. This indicates a transferability of the findings that reach across country- and continent boundaries. Moreover, Bardosh and colleagues [45] also found that implementing ethnography into their study allowed them to assess the socio-cultural drivers and to explore long-term strategies for parasite control. From a methodological point of view, this corroborates with the findings of the present study on the importance of culture and traditions in cooking practice. The inclusion of ethnography into traditional epidemiology is furthermore in line with the One Health framework suggested by [46] and WHO's call for a stronger focus on social determinants in public health research [47].

The study contributes to the global health discourse through identifying potential transmission situations of *T. solium* taeniosis. In demonstrating that transmission may arise not only from a lack of knowledge, but from the local cultural *context* and from the cooks' actions during cooking and their tacit knowledge of how to handle the meat. The thematic analysis showed that there were several issues involved in whether the situations in restaurants or at home could lead to become *actual* potential transmission risks or would remain *potential*.

The key to health promotion interventions must be to address this tacit and embodied knowledge to encourage simple, but effective, changes to the cultural practices. These interventions must be based on relational and detailed understandings of the CoP studied and thus include behavioural changes among cooks in the community (both restaurant and home cooking) in order to ensure a lasting effect on transmission. Social epidemiology and its means of capturing important details on the social determinants of health [48, 49] should be incorporated in future studies. A greater focus on identifying and considering the contextual factors of the research area is important in the assessment of potential influence on (and success of) intervention project outcomes [50].

In conclusion, it appears that the economic, traditional, and cultural settings surrounding a pork cook influence the potential risk of transmission. There is a complex and intricate set of traditional and cultural practices, which are all in turn influenced by the current economic situation of the pork cook. These practices should be taken into account in developing future interventions, from a global health promotion perspective and indeed from a One Health perspective.

## Acknowledgments

The authors would like to thank the participants in this study, particularly the home cooks and restaurant cooks in the four study villages. Furthermore, we wish to thank the village chairmen and helpful villagers in the four villages, as well as the regional and district authorities for assistance in the research and obtaining of research clearances.

Associate professor Ninna Meier of Aalborg University, Denmark, is acknowledged for excellent scientific support in qualitative studies. Ms. Fariji Cannon of Mbeya, Tanzania, is acknowledged for her outstanding translation and research assistance.

## Author Contributions

**Conceptualization:** Karen Schou Møller, Pascal Magnussen, Helena Ngowi, Jeanette Magne.

**Data curation:** Karen Schou Møller.

**Formal analysis:** Karen Schou Møller.

**Funding acquisition:** Stig Milan Thamsborg, Sarah Gabriël.

**Investigation:** Karen Schou Møller.

**Methodology:** Karen Schou Møller, Jeanette Magne.

**Project administration:** Karen Schou Møller.

**Resources:** Karen Schou Møller.

**Supervision:** Pascal Magnussen, Stig Milan Thamsborg, Helena Ngowi, Jeanette Magne.

**Validation:** Pascal Magnussen, Stig Milan Thamsborg, Sarah Gabriël, Helena Ngowi.

**Visualization:** Karen Schou Møller, Jeanette Magne.

**Writing – original draft:** Karen Schou Møller.

**Writing – review & editing:** Karen Schou Møller, Pascal Magnussen, Stig Milan Thamsborg, Sarah Gabriël, Helena Ngowi, Jeanette Magne.

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
