## [Decision Letter · Decision Letter 0]

22 Apr 2022

PONE-D-21-34093The role of cooking practices in the transmission of the foodborne parasite Taenia solium: a qualitative study in an endemic area of Southern TanzaniaPLOS ONE

Dear Dr. Thamsborg,

Thank you for submitting your manuscript to PLOS ONE. After careful consideration, we feel that it has merit but does not fully meet PLOS ONE’s publication criteria as it currently stands. Therefore, we invite you to submit a revised version of the manuscript that addresses the points raised during the review process.

There was some disagreement amongst the reviewers on the value of the approach taken in this manuscript.  We also acknowledge the extraordinary length of time it has taken to have this reviewed.  None the less, the authors are encouraged to respond to the reviews so that the journal can take a second look at the manuscript.

We look forward to receiving your revised manuscript.

Kind regards,

Eric Fèvre

Academic Editor

PLOS ONE

“Grant recipient: SG

This work was funded by the European & Developing Countries Clinical Trials Partnership (EDCTP; grant number DRIA2014-308 SOLID) and the German Federal Ministry of Education and Research (BMBF; grant number: 01KA1617) within the research grant “Evaluation of an antibody detecting point-of-care test for the diagnosis of T. solium taeniosis and (neuro)cysticercosis in communities and primary care settings of highly endemic, resource-poor areas in Tanzania and Zambia, including training of – and technology transfer to the Regional Reference Laboratory and health centers (SOLID)”.

“This work was funded by the European & Developing Countries Clinical Trials Partnership (EDCTP; grant number DRIA2014-308 SOLID) and the German Federal Ministry of Education and Research (BMBF; grant number: 01KA1617) within the research grant “Evaluation of an antibody detecting point-of-care test for the diagnosis of *T. solium* taeniosis and (neuro)cysticercosis in communities and primary care settings of highly endemic, resource-poor areas in Tanzania and Zambia, including training of – and technology transfer to the Regional Reference Laboratory and health centers (SOLID)”.

“Grant recipient: SG

This work was funded by the European & Developing Countries Clinical Trials Partnership (EDCTP; grant number DRIA2014-308 SOLID) and the German Federal Ministry of Education and Research (BMBF; grant number: 01KA1617) within the research grant “Evaluation of an antibody detecting point-of-care test for the diagnosis of T. solium taeniosis and (neuro)cysticercosis in communities and primary care settings of highly endemic, resource-poor areas in Tanzania and Zambia, including training of – and technology transfer to the Regional Reference Laboratory and health centers (SOLID)”.

6. Please amend your manuscript to include your abstract after the title page.

Reviewers' comments:

Reviewer's Responses to Questions

**Comments to the Author**

1. Is the manuscript technically sound, and do the data support the conclusions?

Reviewer #1: Yes

Reviewer #2: Partly

2. Has the statistical analysis been performed appropriately and rigorously? 

Reviewer #1: Yes

Reviewer #2: N/A

3. Have the authors made all data underlying the findings in their manuscript fully available?

Reviewer #1: Yes

Reviewer #2: No

4. Is the manuscript presented in an intelligible fashion and written in standard English?

Reviewer #1: Yes

Reviewer #2: No

5. Review Comments to the Author

Reviewer #1: The paper provides important information on aspects which need to be well understood before we can achieve elimination of the parasite through change of practice. The paper is well written and should be considered for publication after addressing the comments raised.

Reviewer #2: General comment

This manuscript describes cooking and consuming pork in the real-life situation in Tanzania. The authors stated that they follow constructivist, interpretivist approaches to describe the behaviors and perceptions. As it was written so, I read Schwandt 1994, and found it interesting. However, the manuscript is I think not qualitatively analyzing the findings from the views of this constructivist, interpretivist concept in a constructive manner. The manuscript has important information so that readers understand how infection with taenia solium can occur in the daily life. But the writing style is not sharp, and I could not be convinced the value to be published in this journal.

Specific comments

<introduction>

Page 2, line 36: Please consider giving a space after the bracket and ‘-’ after (neuro’-‘)

Page 5, line 104: Loscher and Splitter? Throughout the manuscript.

Page 6, line 124: Selling cooked pork?

Page 7, line 145: please change period to comma for the third digit.

Page 12, line 244: Full-stop please.

Page 13, lines 269-270: Please at least provide percentages. Table 4, please provide percentages at least in Total.

Page 14, Table 5. Please provide percentages within brackets for yes with correct knowledge and yes but do not know what happens.

Page 17, line 346: had been a pig sounds odd.

Page 38. Reference 32. Any book name? Isn’t it a chapter of a book?</introduction>

6. PLOS authors have the option to publish the peer review history of their article (what does this mean?). If published, this will include your full peer review and any attached files.

Reviewer #1: **Yes: **Nicholas Ngwili

Reviewer #2: No

---

## [Author Response · Author response to Decision Letter 0]

30 May 2022

Reviewer #1 comments

"The paper provides important information on aspects which need to be well understood before we can achieve elimination of the parasite through change of practice. The paper is well written and should be considered for publication after addressing the comments raised."

 Many thanks.

"Overall, the abstract there is no mentioned of the different communities of practice which seems to be the basis of this paper. I feel the authors should consider adding a sentence on this aspect in the abstract.

We agree on this and have added a description of the different communities of practice that indeed is essential to the manuscript (Line 36-38)." 

"The discussion is unnecessarily long with some overlaps, the authors should consider making it more precise to shorten it." 

Regarding the reviewer’s comment on the discussion, we have revised and shortened it considerably (from previous 9 pages to current 7 pages). Overlaps have been thoroughly considered and revised accordingly, and sections have been merged and re-written to abridge the text. We hope the reviewers now will find it concise and to the point. 

"The authors should consider also comparing their finding with other studies particularly from Zambia (https://pubmed.ncbi.nlm.nih.gov/27369573/), Uganda (https://www.sciencedirect.com/science/article/pii/S2211912418300968) and (https://www.frontiersin.org/articles/10.3389/fvets.2022.833721/full)".

We want to thank for drawing attention to some of the very latest insight on this subject – the paper from 2022 titled “Stakeholders' Knowledge, Attitude, and Perceptions on the Control of Taenia solium in Kamuli and Hoima Districts, Uganda” by Ngwili et al. When our manuscript was submitted in October of 2021, the authors were not familiar with this very relevant reference. We have included all three mentioned references to support background and for comparison to the present study.

Specific Comments; reviewer #1

Line 73: “It is an unclear what the authors mean by the term “Social situations””. 

The word “social” was deleted (line 104), as it did not add to the understanding, and we now find the wording appropriate.

Line 123 to 124: “How did the researchers ensure that the women they spend time with did not change their practices because they knew they were being watched. How does the authors take care of this in their analysis and conclusions?”

A section on this matter has been added (Lines 906-918) and it now reads: “When applying the ethnographic methods of observing, the researcher must always keep in mind that we can only observe the informant in a somewhat simulated situation. This means that the practices of the cooks during the observation session might be subjected to change because the cook is being observed. This is called the Hawthorne Effect and is a well-known bias of observation (Chen, Weg, Hofmann, & Reisinger, 2015; Jones, 1992). It must therefore be questioned if the situations of the cooking sessions fully mimicked the actual situation of an everyday cooking session in terms of time elapsing. While the home cooks were only observed once for a day while cooking one meal, the restaurant cooks were observed cooking several meals, thus probably making the latter observed practices truer to the actual practices when not under observation. The observed practices might have been altered to increase the social desirability of the practice, and thus the practices that we did observe might represent the perceived desired minimal hygiene levels and safe cooking practices carried out. The Hawthorne effect might have masked several other not observed practices of enhanced transmission risk, but this needs further research.” 

Line 134: “The authors mention that the home cooks pass knowledge to female children. Why female children only?. In many African communities household chores including cooking is shared among the children without regard to their sex unless it was strictly not the case in the study community. In that case that needs to be clarified.”

We agree and the word “female” has been deleted (Line 175). However, the authors did not observe any male children participating in the cooking practices within the community of practice of home cooks in any of the four villages. 

Line 145: “The points in the figures e.g 35.954 amy be taken as decimal points but I believe they are commas to denote thousands?”(sic)

 Indeed. This has been rectified. 

Line 157 to 159: “The naming of the villages may bring some confusion. One may think they have different practices due to their different geographical or physical context which seems to be implied by the names. The author should consider adding a statement to clarify this and make it clear from start.”

In the first mentioning of the anonymous names of the villages, it is added that the naming is unrelated to any common practice in the villages (Lines 214-215). 

Line 161: “What defines a professional cook. This reference is misleading as it may mean cook who have undergo a formal training on cooking. A different term should be considered.” 

We have changed the term to “restaurant cooks” throughout the manuscript to delineate the term to the corresponding term “home cook”, denoting the location of the place of the cooking practice in question. 

Line 337 to 338: “The conclusion on that sentence may not be correct. There could be other factors which may make the cooks not adopt certain practices but not necessarily the inappropriateness of their knowledge. Consider revising.”

The lines have been revised and the term “appropriate” has been deleted (line 435). Furthermore, a sentence on the confidence of the knowledge has been added. 

Line 630: “The professional cook as you call them may have also attained their knowledge from home just like the home cooks long before they ventured into the cooking business. This need to be clarified.”

This has now been changed into including a sentence (lines 729-731) on how none of the restaurant pork cooks mentioned learning how to cook the fried pork sold at restaurants from their mothers at home. This could be due to the fact that the restaurant cooks use the modern recipe of fried pork, where traditional pork recipes include boiled pork. We hope this will clarify the origin of learned recipe. 

Line 739 to 743: “The explanation given as to why the researchers relied on the DVO to select some of the villages is not sufficient. Most officers in Africa no the importance of avoid bias in selection of study participants or study sites. Explaining the need for randomization to the officers would have made them understand and allow the researchers to follow the study procedures in selecting the study sites. This form of bias would have been avoided.”

We fully acknowledge this. However, we do not think that this has influenced the validity of findings in any way. The following has been added to the manuscript in the discussion (lines 954-960): “However, as the present study was of a qualitative nature, randomisation of the study villages was not essential to the study outcome. The most important features of the villages was the consumption of pork, the willingness to participate and share knowledge and practices, and the year-round accessibility by car. After careful explanation of the study to the Regional Veterinary Officer, he assisted by naming two villages, that would fulfill these features, and thus it was decided to include them, as long as the stipulated features were fulfilled.”

Reviewer #2: General comments

This manuscript describes cooking and consuming pork in the real-life situation in Tanzania. The authors stated that they follow constructivist, interpretivist approaches to describe the behaviors and perceptions. As it was written so, I read Schwandt 1994, and found it interesting. However, the manuscript is I think not qualitatively analyzing the findings from the views of this constructivist, interpretivist concept in a constructive manner. The manuscript has important information so that readers understand how infection with taenia solium can occur in the daily life. But the writing style is not sharp, and I could not be convinced the value to be published in this journal.

We expect that all the present revisions will make the paper more “sharp”, concise and spot-on. Further, we find that we have used the outlined methodology in an appropriate manner and cannot fully understand the comment. However, a small section has been added (lines 185-194), describing the interpretive-constructionist paradigm and included how we have used this practically in the data generation. It describes our positioning in the field when observing and partaking in the cooking sessions. We feel that this adds to the understanding of the researcher’s positioning in the field and explains how we have used this positioning to generate data that helps us answer the research question. We hope that the reviewers agree to this. 

The added section reads: “The research was therefore not neutral, because the researcher cannot be a neutral figure in the observation, but must interpret what is observed in practice (Yin, 2016). This non-neutrality begins when electing what cases to include and choosing what subjects to elaborate on in the interviews. Within the constructivist approach, meanings are constructed through experience and through the use of material resources (Guba, 1990). The reconstruction of the practices of cooking a pork meal as a joined experience between researcher and cook allowed for the researcher to observe the cooks’ creation of meaning within the practice of real-time cooking of the pork meal”.

Specific comments; Reviewer #2

Page 2, line 36: “Please consider giving a space after the bracket and ‘-’ after (neuro’-‘)”.

We have revised the sentence and it now reads: 

“Humans ingesting eggs may develop cysticercosis, in particular neurocysticercosis, which can cause severe, chronic headache and epilepsy”

Page 5, line 104: “Loscher and Splitter? Throughout the manuscript”. 

The reference includes the three authors, hence Loscher et al.”

Page 6, line 124: “Selling cooked pork”? 

The study investigates both the cooking of pork and the selling of both raw and cooked pork, so we have opted to keep the wording “cooking and selling of pork”.

Page 7, line 145: “please change period to comma for the third digit”.

Corrected accordingly. 

Page 12, line 244: “Full-stop please”.

Indeed. This has been added. 

Page 13, lines 269-270: “Please at least provide percentages. Table 4, please provide percentages at least in Total”.

Percentages has been added to the lines as well as to Table 4 in Total. 

Page 14, Table 5. “Please provide percentages within brackets for yes with correct knowledge and yes but do not know what happens”.

This has been added to Table 5.

Page 17, line 346: “had been a pig sounds odd”.

The word “available” has been added to clarify the meaning of the sentence. 

Page 38. Reference 32. “Any book name? Isn’t it a chapter of a book?”

No, this is a document from 1993 by the Tanzanian Government on the meat inspection of animals before and after slaughter. We have not managed to find it online. The document was sent to one author personally. This has now been clarified in the reference list. 

Chen, L., Weg, M., Hofmann, D., & Reisinger, H. (2015). The Hawthorne Effect in Infection Prevention and Epidemiology. Infection control and hospital epidemiology, 36, 1-7. doi:10.1017/ice.2015.216

Guba, E. G. (1990). The paradigm dialog. Newbury Park, Calif: Sage.

Jones, S. R. G. (1992). Was There a Hawthorne Effect? The American journal of sociology, 98(3), 451-468. doi:10.1086/230046

Yin, R. K. (2016). Qualitative research from start to finish (2nd ed. ed.). New York: Guilford Press.

---

## [Editor Report · Decision Letter 1]

7 Sep 2022

The role of cooking practices in the transmission of the foodborne parasite Taenia solium: a qualitative study in an endemic area of Southern Tanzania

PONE-D-21-34093R1

Dear Dr. Thamsborg,

We’re pleased to inform you that your manuscript has been judged scientifically suitable for publication and will be formally accepted for publication once it meets all outstanding technical requirements.

Kind regards,

Eric Fèvre

Academic Editor

PLOS ONE

Additional Editor Comments (optional):

We apologise for the time taken to secure a final decision on this manuscript. The comments returned to the major revision are now considered acceptable for this manuscript to be accepted by the journal.
---

## [Editor Report · Acceptance letter]

26 Sep 2022

PONE-D-21-34093R1 

The role of cooking practices in the transmission of the foodborne parasite *Taenia solium*: a qualitative study in an endemic area of Southern Tanzania 

Dear Dr. Thamsborg:

I'm pleased to inform you that your manuscript has been deemed suitable for publication in PLOS ONE. Congratulations! Your manuscript is now with our production department. 

Kind regards, 

on behalf of

Prof. Eric Fèvre 

Academic Editor

PLOS ONE